# TAR-TVG: Enhancing LVLMs with Timestamp Anchor-Constrained Reasoning for Temporal Video Grounding

## Abstract

Temporal video grounding aims to localize relevant video segments based on a given query. Large Vision-Language Models (LVLMs) can address this by taking a video and query as input and outputting the time duration. Recently, some methods fine-tune LVLMs with reinforcement learning (RL), encouraging them to generate reasoning traces for better interpretability. They also prompt the model to include `<timestamp></timestamp>` tags into the reasoning process to strengthen the connection between the reasoning and the final output. However, these prompts only implicitly guide the model to output timestamp tags, often leading to missing, incorrect-formatted, or irrelevant tags. To address this issue, we propose **T**imestamp **A**nchor-constrained **R**easoning for **T**emporal **V**ideo **G**rounding (TAR-TVG). By designing reinforcement learning reward functions, we explicitly enforce the inclusion of timestamp tags as anchors within the reasoning traces, providing explicit format control and accuracy validation based on soft IoU. Furthermore, when multiple timestamp anchors appear, the reward function is designed to ensure that the accuracy of these anchors progressively improves, thereby mimicking the human-like thought process of refining from coarse to fine. These additional constraints on timestamp anchors encourage the model to better understand the task of temporal video grounding, thereby improving its grounding performance. Additionally, we first run an RL stage purely for data collection. The collected samples are then used to SFT a fresh base model, and we finally apply RL fine-tuning to the SFT-initialized model. Experiments show that our model achieves state-of-the-art performance while producing verifiable reasoning chains with progressively refined temporal estimations.

## 1 Introduction

Temporal Video Grounding (TVG) is a fundamental task in video understanding that requires models to precisely localize temporal segments in untrimmed videos corresponding to natural language queries (Anne Hendricks et al., 2017; Gao et al., 2017; Krishna et al., 2017; Lin et al., 2023; Zhang et al., 2023). Imaging in smart home environments, TVG can enable AI assistants to accurately locate moments like "when the child entered the kitchen" from hours of footage (Yang et al., 2025; Grauman et al., 2022; Sigurdsson et al., 2016). This capability is critical for numerous real-world applications, including video surveillance (Tian et al., 2009), intelligent video retrieval (Anne Hendricks et al., 2017; Caba Heilbron et al., 2015), and human-computer interaction systems (Zhou et al., 2023).

Previous TVG methods can be broadly categorized into three paradigms: (1) Vision Language Pretraining (VLP) approaches leverage pretrained visual encoders in combination with specialized feature fusion and grounding modules (Jang et al., 2023; Mu et al., 2024; Zhu et al., 2023; Zhang et al., 2021b; Zeng et al., 2020). Although effective, these methods suffer from error accumulation due to disjoint feature extraction and grounding steps. (2) Recent methods use LVLMs to directly predict segment boundaries. (Qu et al., 2024; Ren et al., 2024; Huang et al., 2024; Zeng et al., 2025), bypassing any intermediate reasoning processes and therefore lacking interpretability. (3) Reasoning-Enhanced Models, such as works like Time-R1 (Wang et al., 2025b)

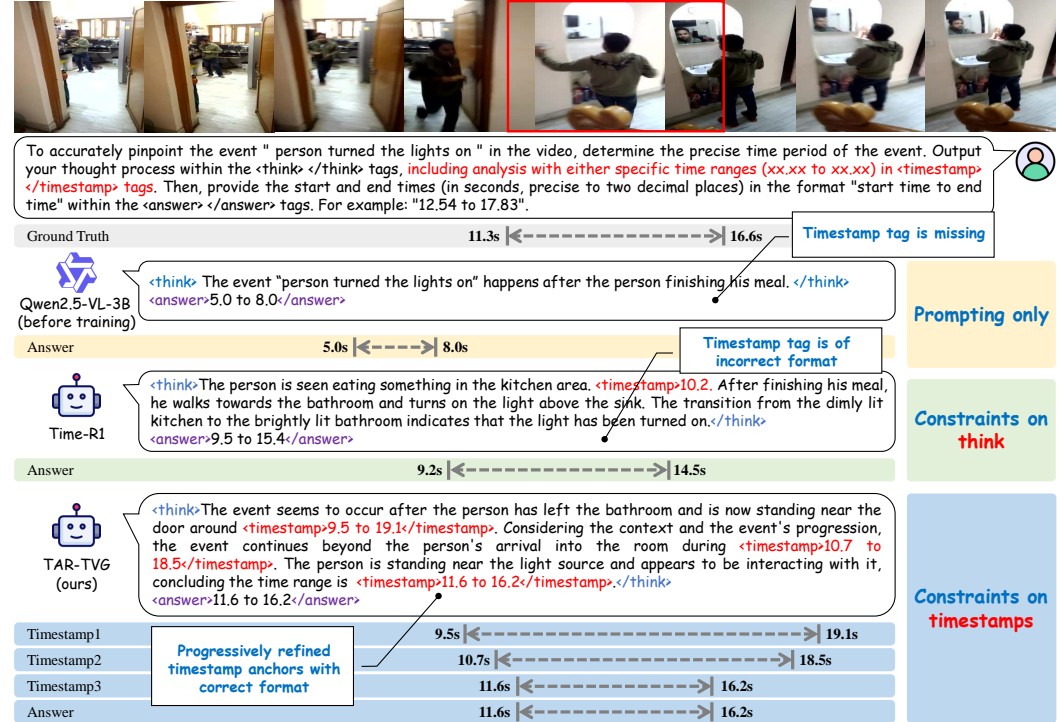

Figure 1: Differences between Qwen2.5-VL, Time-R1, and the proposed TAR-TVG. Qwen2.5-VL is simply prompted to generate a reasoning trace that includes timestamp tags. Time-R1 goes further by enforcing constraints on the format of the think content. However, neither method explicitly constrains the timestamp tags themselves. In contrast, our method, TAR-TVG, directly imposes explicit constraints on the timestamp tags, regulating their format, quantity, and accuracy, and requires that later timestamps become increasingly precise, enabling a progressive refinement of temporal reasoning.

and VideoChat-R1 (Li et al., 2025), inspired by DeepSeek-R1 (Guo et al., 2025), adopt chain-of-thought prompting where models first generate `<think>...</think>` reasoning chains before producing `<answer>...</answer>` tags.

A key insight is that the reasoning content within the *think* tag should be meaningfully associated with the result in the *answer* tag, so that the reasoning process can effectively lead to the correct output. Time-R1 proposes using prompts (see Figure 1) to encourage the model to include timestamp tags within the *think* tag, where these tags specify time ranges and are meant to link the reasoning process to the final output in the *answer* tag. However, this approach relies solely on prompts to induce the model to include the tags which is an implicit mechanism that the model may not follow. As a result, the timestamp tags may be entirely missing, or their formats may be incorrect (as shown in Figure 1). Moreover, there is no explicit constraint imposed on the timestamp tags, preventing their significant potential from being fully leveraged.

Based on these considerations, we propose Timestamp Anchor-constrained Reasoning for Temporal Video Grounding (TAR-TVG). By designing new RL reward functions, we explicitly require the model to output timestamp tags in the think content. We impose constraints on both the number and format of these tags, thereby increasing their likelihood of appearing correctly in the reasoning trace. At the same time, we treat these timestamp tags as anchors, and propose using soft IoU to evaluate their accuracy. Most importantly, with these anchors in place, we can enforce that later anchors are more accurate than earlier ones. This enables a progressive refinement of timestamp accuracy throughout the reasoning process, mimicking the human-like thought pattern of refining from coarse to fine. Through these additional constraints, we argue that if the model is able to satisfy them, it achieves a better understanding of the temporal grounding task, which ultimately leads to the improvement in grounding accuracy.

Another problem is that baseline models such as Qwen2.5-VL-3B/7B struggle to generate reasoning traces with valid timestamp tags initially, even when explicitly prompted. Our experiments reveal that in 76% cases, these models either fail to insert timestamp tags entirely or produce some with wrong format. Without tags, the proposed anchor-based constraints become inapplicable, thus degrading the training efficiency.

To address this, we introduce a data collection strategy during 7B model training that distinguishes our work from prior methods relying on direct distillation from large-scale models. Specifically, we first employ reinforcement learning to train a temporary 7B model(Qwen2.5-VL-7B) whose generation quality progressively improves over time. From this process, we filter and select high-quality outputs, particularly those with accurate timestamp tags, to curate a 30K Chain-of-Thought dataset. Crucially, this temporary model is utilized solely for data synthesis rather than downstream training.

The curated dataset is then used to fine-tune (SFT) the origin model to improve its ability to generate timestamp-aligned reasoning. Finally, the SFT-enhanced model undergoes further refinement using RL with anchor-constraint rewards, leveraging its improved initialization.

In summary, the key contributions of this paper are: (1) We propose TAR-TVG, a new reinforcement learning method that introduces timestamp anchors into the model's reasoning process. These anchors enable step-by-step verification and enforce progressive refinement. This approach effectively guides the model to better understand temporal grounding in videos. (2) We present an efficient data collection strategy that automatically extracts high-quality reasoning traces during model training, eliminating the need for manual dataset creation. (3) Extensive experiments demonstrate the superiority of our method. In Charades-STA, TAR-TVG achieves a new state-of-the-art performance, with the best mIoU (61.1) and the highest R1@0.7 (50.2), outperforming existing approaches.

## 2 RELATED WORKS

**Temporal Video Grounding.** Temporal Video Grounding (TVG) (Gao et al., 2017; Krishna et al., 2017) aims to precisely locate video segments corresponding to text queries. Early methods mainly rely on Vision-Language Pre-training (VLP) models (Zhang et al., 2021a; Lei, 2021; Lin et al., 2023; Cao et al., 2025), while subsequent approaches leverage large language models (LLMs) with fine-tuned visual encoders or open-source large vision-language models (Qu et al., 2024; Huang et al., 2024; Zeng et al., 2025; Ren et al., 2024), though both lack interpretability and temporal awareness. Recently, reasoning and reinforcement learning (RL) have been introduced into LLMs (Jin et al., 2025; Wu et al., 2025), with Time-R1 (Wang et al., 2025b) achieving state-of-the-art grounding performance through a reasoning-driven LVLM and VideoChat-R1 (Li et al., 2025) extending to broader spatio-temporal tasks, yet these RL-based methods still lack explicit constraints on the reasoning process, which our method addresses.

**Large Vision-Language models Reasoning and Reward Optimization.** Recent advances in reasoning with large language models (LLMs) (Achiam et al., 2023) have demonstrated outstanding performance in mathematics (Liu et al., 2024) and coding (Gao et al., 2024) tasks, as exemplified by models such as OpenAI's o1 and DeepSeek's R1 (Guo et al., 2025). Subsequent efforts (Zhang et al., 2025a;b) have extended reasoning capabilities to the visual domain with LVLMs, initially focusing on images. More recently, increasing attention has been paid to reasoning in videos. For instance, TVG-R1 (Chen et al., 2025a) is specifically designed for temporal video grounding, whereas Video-R1 (Feng et al., 2025) advances video reinforcement learning with T-GRPO, and Long-RL (Chen et al., 2025b) addresses the challenge of reasoning over long videos. All of these methods are based on task-specific metrics as rewards and use rule-based approaches. Some methods also employ additional reward models (Wang et al., 2025a) which evaluates the reasoning process of the model's output. However, such rule-based approaches often result in a disconnect between the reasoning process and the final answer, while purely model-based rewards can be computationally expensive, especially when the rewards need to be assigned to the content of the reasoning itself. To strike a balance, we adopt a rule-based method that rewards intermediate steps by introducing timestamp during the thinking process as intermediate verification points, effectively combining the advantages of both approaches.

## 3  METHOD

Given a video $V$, Temporal Video Grounding (TVG) can be formulated as $[\hat{t}_s, \hat{t}_e] = f(V, p, q)$, where $[\hat{t}_s, \hat{t}_e]$ denotes the predicted start and end boundaries of the target segment, $p$ is the used prompt and $q$ is the original query. The temporal segment of ground truth is represented as $[t_s, t_e]$.

Figure 2 illustrates our reinforcement learning framework employing the GRPO algorithm. Given video, query and prompt inputs, the Large Vision-Language Model (LVLM, specifically Qwen2.5-VL-3B/7B in this work) is prompted to generate structured outputs comprising: (1) temporal boundary predictions $\hat{t}_s, \hat{t}_e$ enclosed in `<answer>...</answer>` tags, and (2) reasoning traces within `<think>...</think>` block containing intermediate `<timestamp>...</timestamp>` predictions. For these outputs, we compute three reward functions: (1) Format Reward ensures to output correct think, answer, and timestamp tags, (2) Soft IoU Reward measures temporal overlap between predicted timestamp/answer tags and ground truth, and (3) Timestamp Anchor-Constrained Reward (our core innovation) constrains the reasoning process by requiring progressive refined `<timestamp>` predictions within `<think>` blocks.

The subsequent text delineates our proposed reward functions: the Format Reward, Soft IoU Reward, and Timestamp Anchor-Constrained Reward. Finally, we outline our strategy for data collection.

### 3.1  FORMAT REWARD

We prompt (see the prompt in the appendix B.4) the model to output in the following structure: `<think>...<timestamp>...</timestamp>...<timestamp>...</timestamp>...` `</think> <answer>...</answer>`.

In our `<think>` block, the model first provides an initial analysis of the video, then outputs a preliminary answer in the first `<timestamp>` block. It then continues to analyze this initial answer in combination with its earlier reasoning and the video content, producing a refined or corrected answer in the second `<timestamp>` block, and so on until it wants to stop. In the `<answer>` block, the model provides its final answer, which we set as the same in the last timestamp tag in the think block. We use regular expressions to verify whether the model's output conforms to this required format. If the format is correct, we give a score of 3 to the model, otherwise 0:

$$r_{\text{Format}}(o) = \begin{cases} 0, \text{if } o \text{ does not match format} \\ 3, \text{if } o \text{ matches format} \end{cases} \quad (1)$$

### 3.2  SOFT IOU REWARD

In previous work, they usually adopt the following IoU reward:

$$r_{\text{IoU}}(o) = \frac{\max\left(0, \min(t_e, \hat{t}_e) - \max(t_s, \hat{t}_s)\right)}{\max(t_e, \hat{t}_e) - \min(t_s, \hat{t}_s)}. \quad (2)$$

In this paper, we propose a slightly different soft IoU to compute the reward for the model's output:

$$r_{\text{sIoU}}(o) = \frac{\min(t_e, \hat{t}_e) - \max(t_s, \hat{t}_s)}{\max(t_e, \hat{t}_e) - \min(t_s, \hat{t}_s)}. \quad (3)$$

The only difference is that the molecule has no lower bound of zero. Soft IoU can measure the distance between the predicted and ground-truth segments even when there is no overlap which the previous IoU cannot achieve, enabling stable training. For example, in the early stages of training, if the model predicts a segment from 15.4s to 18.0s while the ground truth is from 3.4s to 12.2s, soft IoU yields a score of -0.21, whereas standard IoU gives 0. As a result, the standard IoU is not sensitive to the degree of misalignment in the predictions, leading to slower convergence.

### 3.3  TIMESTAMP ANCHOR-CONSTRAINED REWARD

Besides the format and soft IoU reward functions, we further design a timestamp anchor-constrained reward to apply constraints to the timestamp anchors.

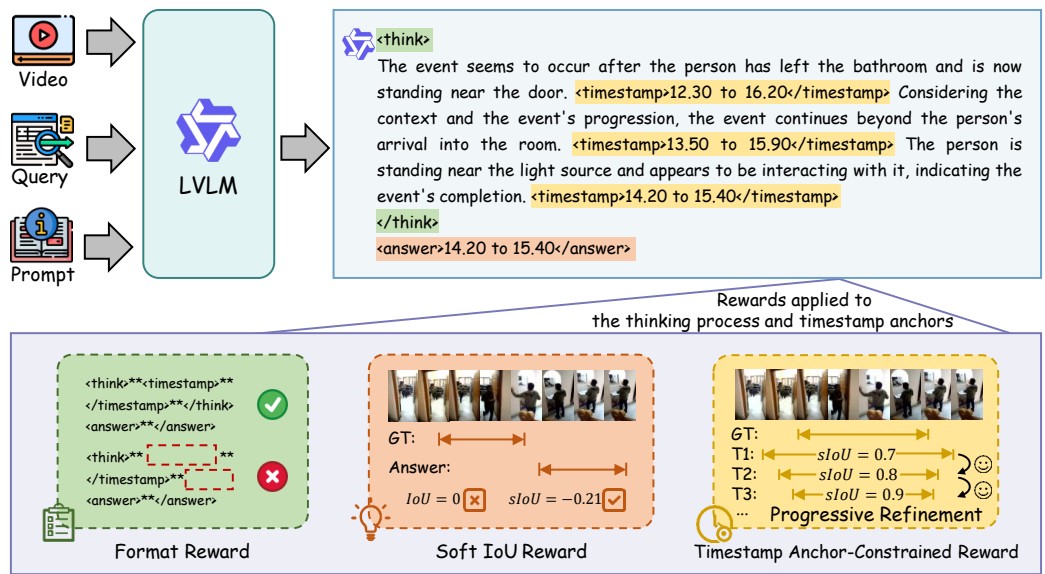

Figure 2: Overview of our proposed Timestamp Anchor-constrained Reasoning for Temporal Video Grounding (TAR-TVG) method. We apply a format reward, a soft IoU reward, and a timestamp anchor-constrained reward to the think, answer and timestamp tags generated by the LVLM.

First, we find format-correct timestamp tags in the think content, and extract the time duration in-between them. Let $\hat{s}$ be the number of timestamp tags generated by the model. We compute the weighted sum of the soft IoUs between the time ranges in `<timestamp>` tags and the ground truth:

$$\text{TAR}_{\text{sIoU}}(o) = \sum_{i=1}^{\hat{s}} i \, \text{sIoU}_i(o), \qquad (4)$$

where $\text{sIoU}_i(o)$ computes the soft IoU score for the $i^{th}$ timestamp anchor of output $o$, and $i$ is the weight, with more recent timestamp anchors having greater weights, enforcing them to be more accurate.

Second, we explicitly require that later timestamp anchor be more accurate than earlier ones, achieving progressive refinement. The progressive refinement reward for the $i^{th}$ timestamp anchor of output $o$ can be formalized as:

$$\delta_i(o) = \begin{cases} 1 & \text{if } \text{sIoU}_i(o) > \text{sIoU}_{i-1}(o), \\ -1 & \text{otherwise.} \end{cases} \qquad (5)$$

Based on this, $\text{TAR}_{\text{refine}}$ is defined as:

$$\text{TAR}_{\text{refine}}(o) = \sum_{i=2}^{\hat{s}} \delta_i(o). \qquad (6)$$

Finally, we find that $\hat{s}$ is not the larger the better. If the size of $\hat{s}$ is not limited, it may grow large, which increases the difficulty of achieving progressive refinement and can ultimately degrade the model's training performance. Therefore, we impose a penalty to $\hat{s}$. Let $s$ be the target number of `<timestamp>` tags. We use the following reward to prevent $\hat{s}$ from deviating from $s$ too much.

$$\text{TAR}_{\text{num}}(o) = (\hat{s} - s)^2. \qquad (7)$$

Finally, our timestamp anchor constrained reward function is defined as:

$$r_{\text{TAR}}(o) = \text{TAR}_{\text{sIoU}}(o) + \beta \, \text{TAR}_{\text{refine}}(o) - \gamma \, \text{TAR}_{\text{num}}(o), \qquad (8)$$

where $\beta$=1, $\gamma$=5 are hyperparameters. See the appendix C for ablation on these hyperparameters. The overall reward of the whole reinforcement learning is formulated as follows.

$$r(o) = r_{\text{Format}}(o) + r_{\text{sIoU}}(o) + r_{\text{TAR}}(o). \qquad (9)$$

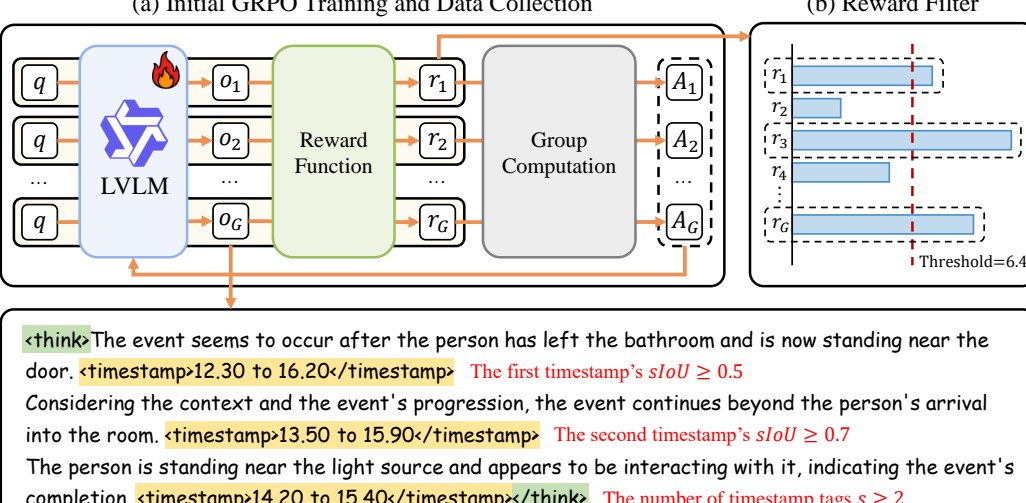

(a) Initial GRPO Training and Data Collection

(b) Reward Filter

<think>The event seems to occur after the person has left the bathroom and is now standing near the door. <timestamp>12.30 to 16.20</timestamp> The first timestamp's *sIoU* ≥ 0.5

Considering the context and the event's progression, the event continues beyond the person's arrival into the room. <timestamp>13.50 to 15.90</timestamp> The second timestamp's *sIoU* ≥ 0.7

The person is standing near the light source and appears to be interacting with it, indicating the event's completion. <timestamp>14.20 to 15.40</timestamp></think> The number of timestamp tags *s* ≥ 2

<answer>14.20 to 15.40</answer>

(c) Chain of Thought Filter

Figure 3: Data Collection Strategy. Unlike prior work that directly distills data from larger pretrained models, we train a small 7B model with GRPO and collect its generated samples. As training progresses, the quality of these samples gradually improves, after which we apply a reward-based filter to obtain high-quality reasoning traces.

## 3.4 DATA COLLECTION

Training the proposed model directly (as illustrated in Figure 2) is non-trivial. Despite extensive experimentation with various prompts to encourage the model to output timestamp tags, it consistently struggles to generate them reliably, particularly when the prompts are complex. This is partly because the baseline model we use (i.e., Qwen2.5-VL-3B/7B) has limited capability in fully comprehending complex prompts. When no timestamp tags are produced, the timestamp anchor-constrained reward cannot be computed, which prevents effective training.

To address this issue, we propose a Data Collection strategy (see Figure 3) involving the training of a 7B model for data acquisition. This stands in contrast to prevailing methods, which typically distill data directly from significantly larger models (e.g. GPT-4o). First, we directly execute the reinforcement learning process, implemented by GRPO. Given a video $V$, a prompt $p$, and a query $q$, the GRPO algorithm generates outputs $o$ and computes a reward $r$ for each, ultimately optimizing the model based on the group advantages $A$. During this process, we accumulate 186K outputs $o$ and filter them down to 30K. The filtering criteria are: $r$ must be greater than 6.4, the number of generated timestamp tags must be at least 2, the sIoU of the first timestamp must exceed 0.5, and the second one must exceed 0.7. Although the probability of satisfying the above requirements is low, some high-quality data can still be generated. This approach allows us to collect a dataset without any manual intervention.

## 4 EXPERIMENTS

### 4.1 EXPERIMENTAL SETUP

**Benchmarks.** We evaluate our model on four video grounding datasets: Charades-STA (Gao et al., 2017) (6,672 indoor activity videos, 30.6s avg, 12,408 train/3,720 test clip-query pairs), QVHighlights (Lei, 2021) (4,855 train/1,020 val video-text pairs that contain only a single segment), ActivityNet-Caption (Caba Heilbron et al., 2015) (37,421 train/17,505 val/17,031 test samples), and TVGBench (Wang et al., 2025b) (800 samples). The last one aggregates part test sets from ActivityNet-Caption, Charades-STA, HiREST (Zala et al., 2023), EgoNLQ (Grauman et al.,

Table 1: Performance of temporal video grounding on Charades-STA, QVHighlights. Methods marked with * are trained with additional TVG datasets (See the Appendix B.1 for their implementation and the dataset they use), while FT with ✓ indicates that the model is fine-tuned on the corresponding Charades-STA or QVHighlights dataset. We compare our method against existing 3B, 7B open-source LVLMs, as well as state-of-the-art VLP models.

| Type | Method | Size | FT | Charades-STA | | | | QVHighlights | | | |
|------|--------|------|----|------|--------|--------|--------|------|--------|--------|--------|
| | | | | mIoU | R1@0.3 | R1@0.5 | R1@0.7 | mIoU | R1@0.3 | R1@0.5 | R1@0.7 |
| VLP | 2D-TAN (Zhang et al., 2020) | - | ✓ | - | 57.3 | 45.8 | 27.9 | - | - | - | - |
| | M-DETR (Lei, 2021) | - | ✓ | 45.5 | 65.8 | 52.1 | 30.6 | - | - | 53.9 | 34.8 |
| | UniVTG (Lin et al., 2023) | - | ✓ | 50.1 | 70.8 | 58.1 | 35.6 | - | - | 59.7 | - |
| | EaTR (Jang et al., 2023) | - | ✓ | - | - | 68.4 | 44.9 | - | - | 61.4 | 45.8 |
| | CG-DETR (Moon et al., 2023) | - | ✓ | 50.1 | 70.4 | 58.4 | 36.3 | - | - | 67.4 | 52.1 |
| | FlashVTG (Cao et al., 2025) | - | ✓ | - | - | 70.3 | 49.8 | - | - | 73.1 | 57.3 |
| SFT | ChatVTG (Qu et al., 2024) | 7B | | - | 52.7 | 33.0 | 15.9 | - | - | - | - |
| | TimeChat* (Ren et al., 2024) | 7B | ✓ | - | 32.2 | 13.4 | 36.2 | - | - | - | - |
| | VTimeLLM (Huang et al., 2024) | 7B | | - | 51.0 | 27.5 | 11.4 | - | - | - | - |
| | TimeSuite (Zeng et al., 2025) | 7B | | - | 69.9 | 48.7 | 24.0 | - | - | - | - |
| | TRACE* (Guo et al., 2024) | 7B | ✓ | - | - | 40.3 | 19.4 | - | - | - | - |
| | TimeSuite* (Zeng et al., 2025) | 7B | ✓ | - | 79.4 | 67.1 | 43.0 | - | - | - | - |
| RL | Time-R1* (Wang et al., 2025b) | 3B | ✓ | 55.3 | 79.5 | 65.1 | 37.5 | - | - | - | - |
| | TAR-TVG (ours) | 3B | ✓ | **57.0** | **81.0** | 67.2 | **40.6** | - | - | - | - |
| | Time-R1* (Wang et al., 2025b) | 7B | ✓ | 58.8 | 82.8 | 72.2 | 50.1 | 59.1 | 74.1 | 66.2 | 52.7 |
| | TAR-TVG (ours) | 7B | ✓ | **61.1** | **83.6** | 71.4 | **50.2** | 65.9 | 85.6 | 76.1 | 58.5 |

Table 2: **Zero-shot** comparison on the ActivityNet-Captions and TVGBench benchmarks. All models have 7B parameters.

| Method | ActivityNet-Captions | | | | TVGBench | | | |
|--------|------|--------|--------|--------|------|--------|--------|--------|
| | mIoU | R1@0.3 | R1@0.5 | R1@0.7 | mIoU | R1@0.3 | R1@0.5 | R1@0.7 |
| VideoChat (Li et al., 2023) | 7.2 | 8.8 | 3.7 | 1.5 | - | - | - | - |
| Video-ChatGPT (Maaz et al., 2023) | 18.9 | 26.4 | 13.6 | 6.1 | - | - | - | - |
| ChatVTG (Qu et al., 2024) | 27.2 | 40.7 | 22.5 | 9.4 | - | - | - | - |
| TimeChat (Ren et al., 2024) | - | 36.2 | 20.2 | 9.5 | - | 22.4 | 11.9 | 5.3 |
| HawkEye (Wang et al., 2024) | - | 49.1 | 29.3 | 10.7 | - | - | - | - |
| VTime-LLM (Huang et al., 2024) | - | 44.0 | 27.8 | 14.3 | - | - | - | - |
| Videochat-Flash (Li et al., 2024) | - | - | - | - | - | 32.8 | 19.8 | 10.4 |
| TRACE (Guo et al., 2024) | - | - | - | - | - | 37.0 | 25.5 | 14.6 |
| TimeSuite (Zeng et al., 2025) | - | - | - | - | - | 31.1 | 18.0 | 8.9 |
| Time-R1 (Wang et al., 2025b) | - | 58.6 | 39.0 | **21.4** | 29.2 | 41.8 | 29.4 | **16.4** |
| ours(STA) | 37.2 | 55.1 | 35.1 | 17.3 | 29.0 | 42.0 | 28.0 | 15.4 |
| ours(STA+QV) | **41.1** | **61.5** | **39.8** | 19.8 | **30.6** | **42.8** | **31.1** | 16.0 |

2022) and TaCoS (Regneri et al., 2013) as a comprehensive benchmark, evaluating 11 query types with balanced video-query distribution for fair assessment.

**Implementation details.** We use Qwen2.5-VL-3B/7B as our base models. We first generate and filter the 30K CoT data, and then train an SFT model using this data. We then fine-tune the models for 3 epochs on the Charades-STA training set, followed by 1 epoch of training on the QVHighlights training set. The SFT stage for both the 3B and 7B models is conducted using $4 \times$ NVIDIA A100 40GB GPUs and is implemented via LoRA fine-tuning with a rank $r = 16$ and scaling factor $\alpha = 32$. For GRPO training, the 3B model is trained with $8 \times$ A100 40GB GPUs, and the 7B model is trained with $16 \times$ A100 40GB GPUs. More training details are provided in appendix B.2.

## 4.2 EVALUATION

We compare our TAR-TVG model with state-of-the-art TVG methods, including feature-based vision-language pretraining (VLP) models, large vision-language models fine-tuned through supervised fine-tuning (SFT), and recent approaches that leverage reinforcement learning (RL) to generate chain-of-thought reasoning.

**Comparison on Charades-STA.** As shown in Table 1, our TAR-TVG (7B) achieves the highest mIoU of 61.1 and R1@0.3 of 83.6 on the Charades-STA benchmark. This performance is achieved using only the Charades-STA training set, surpassing methods (denoted with *) that rely on additional training data, including but not limited to YT-Temporal (Yang et al., 2023), DiDeMo (Anne Hendricks et al., 2017), QuerYD (Oncescu et al., 2021), InternVid (Wang et al.,

2023), and HowTo100M (Miech et al., 2019). For instance, in terms of R1@0.3, our model outperforms Time-R1 (82.8) trained with an additional 2.5k external samples, as well as all other types of TVG models. Our smaller 3B version of TAR-TVG surpasses most VLP-based models and reaches performance comparable to some existing 7B models.

**Comparison on QVHighlights.** On the QVHighlights dataset, our method significantly outperforms most VLP-based models and reinforcement learning-based approaches (Table. 1). Specifically, we observe improvements of +6.8 mIoU, +9.9 R1@0.5, and +5.8 R1@0.7 compared to Time-R1, demonstrating strong generalization of our method between different TVG scenarios.

**Zero-Shot Comparison on ActivityNet-Captions and TVGBench.** In the zero-shot setting, we further evaluate our method on two challenging benchmarks: ActivityNet-Captions and TVGBench. Table 2 presents the results on ActivityNet-Captions, the experimental results show that our method achieves the highest R1@0.3 score (61.5) and the highest mIoU (41.1) under the zero-shot setting. Similarly, our approach achieves the highest mIoU of 30.6 on TVGBench.

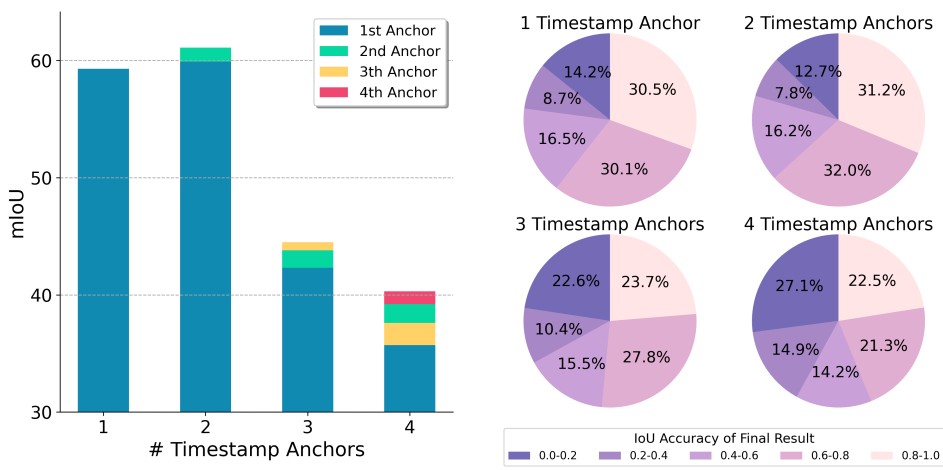

Figure 4: **Left**: mIoU performance with different numbers of timestamp anchors. As the number of anchors increases to 3 or 4, the mIoU of each anchor decreases. **right**: Distribution of the accuracy of the final grounding results with different numbers of timestamps. When there are 2 timestamps, 31.2% of the IoU of the grounding results is distributed between 0.8 and 1.0, which is the best.

Table 3: Ablation on Three-Stage Training Strategy.

| Type | mIoU | R1@0.3 | R1@0.5 | R1@0.7 |
|------|------|--------|--------|--------|
| +SFT | 40.0 | 58.8 | 41.8 | 21.0 |
| +GRPO | 59.3 | 81.5 | 70.1 | 45.9 |
| +SFT + GRPO | **61.1** | **83.6** | **71.4** | **50.2** |

Table 4: Ablation on $TAR_{sIoU}$, $TAR_{num}$ and $TAR_{refine}$ rewards.

| $TAR_{sIoU}$ | $TAR_{refine}$ | $TAR_{num}$ | mIoU | R1@0.5 | R1@0.7 |
|--------------|----------------|-------------|------|--------|--------|
| ✓ | | | 39.1 | 50.2 | 31.2 |
| ✓ | ✓ | | 41.1 | 51.3 | 32.4 |
| ✓ | | ✓ | 58.5 | 70.2 | 47.9 |
| ✓ | ✓ | ✓ | **61.1** | **71.4** | **50.2** |

## 4.3 ABLATION STUDY AND DISCUSSION

We conduct a series of ablation studies on the 7B model.

**Number of Timestamp Anchors.** Figure 4 illustrates the impact of different number of timestamp anchors on model performance. Both the figures on the left and right show that setting the anchor number to 2 yields the best overall performance, surpassing the single-anchor case. However, counter-intuitively, the performance becomes worse when the number of anchors reaches 3 or 4. We believe this is because the progressive refinement constraint is easier to achieve when there are only 2 anchors, whereas as the number of anchors increases, it becomes more difficult to ensure that each subsequent anchor is progressively better, thereby reducing the training effectiveness. We think this may also be related to the inherent capability of the base model. Using a stronger model than Qwen2.5-VL-7B might potentially support a larger number of anchors.

**Impact of Three-Stage Training Strategy.** As shown in Table 3, incorporating the CoT data for supervised fine-tuning substantially enhances the model's temporal grounding ability during subsequent GRPO training. Training with reinforcement learning alone achieves only 45.9 R1@0.7, while applying chain-of-thought (CoT) supervision solely in the SFT stage yields just 21.0 R1@0.7, as SFT mainly focuses on token-level optimization and thus provides limited improvement in temporal understanding. However, when reinforcement learning is further applied to the SFT-trained model, both reasoning ability and video comprehension are strengthened, resulting in the best performance of 50.2 R1@0.7 and 61.1 mIoU.

**Ablation on TAR Rewards.** Table 4 describes the ablation study on TAR rewards, involving $TAR_{sIoU}$, $TAR_{num}$, and $TAR_{refine}$. In all rows, $TAR_{sIoU}$ is always employed because it is a soft IoU constraint applied to timestamp anchors. Omitting it may generate an overly wide time span and only slightly narrow it at each step. $TAR_{num}$ is very useful, since omitting it the model may generate an excessive number of timestamp anchors, resulting in worse performance as shown in the first two rows. With $TAR_{sIoU}$, adding $TAR_{num}$ achieves an R1@0.7 of 47.9. Combining all components reaches the highest 50.2, demonstrating their complementary effect.

**Soft IoU vs Standard IoU.** We compared soft IoU with the standard IoU in Figure 5. In the early stages of training, soft IoU can provide informative feedback even when the predicted and ground-truth segments do not overlap, offering meaningful rewards to guide optimization. As shown in Figure 5, sIoU consistently outperforms standard IoU throughout training and leads to better final performance.

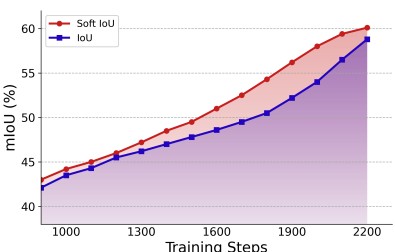

Figure 5: Soft IoU vs Standard IoU.

**About Thinking Length.** We recorded the thinking lengths of models of different sizes. Figure 6 show that the 3B model generally produces longer thinking sequences compared to the 7B model, while 7B model has better grouding performance than 3B. For example, when only GRPO is applied, the 3B model generates on average 11.1 (186.1-175.0) more tokens than the 7B model. Second, we observe that after SFT, all models show improved performance. With SFT, the thinking length of the 3B model remains nearly unchanged. For the 7B model, the thinking length decreases from 175 tokens to 158.4. Both results indicate that stronger models require less thinking to achieve better performance.

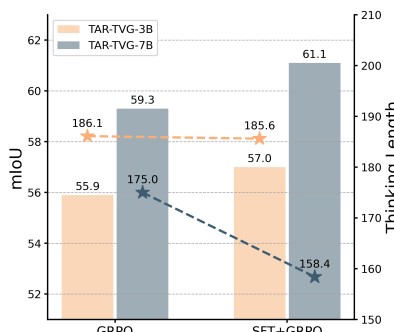

Figure 6: The thinking length of different model size. The bar chart represents the mIoU, while the stars indicate the average number of tokens.

## 5 CONCLUSION

We propose TAR-TVG, a Timestamp Anchor-constrained Reasoning framework for Temporal Video Grounding. To address the limitations of existing methods in supervising the reasoning process, we introduce a progressive, step-by-step timestamp refinement mechanism that leads to increasingly accurate predictions. Our contributions include: (1) TAR-TVG, a reinforcement learning method that introduces timestamp anchors into the reasoning process in a novel way, enabling verification and progressive refinement; (2) an efficient training strategy which can automatically generate high-quality datasets during training; (3) state-of-the-art performance on Charades-STA, along with improved results on other TVG benchmarks. Our method achieves the best reported mIoU of 61.1 on the Charades-STA dataset.

## ETHICS STATEMENT

This work relies exclusively on publicly available open-source models and datasets. No proprietary or sensitive data were used, and the datasets do not contain personally identifiable information. We followed the licenses and usage guidelines associated with these resources. While our research does not introduce direct ethical risks, we acknowledge that, as with many machine learning models, there remains the potential for misuse in unintended contexts. We encourage responsible use and further investigation into mitigating potential biases present in open-source datasets.

## REPRODUCIBILITY STATEMENT

We are committed to ensuring the reproducibility of our work. The models, training datasets, prompts, and hyperparameters used in our experiments are fully documented in Section 4.1 and Appendix B. These descriptions should allow researchers to replicate our experimental setup and results without requiring additional resources beyond those specified.

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

# A   ADDITIONAL ANALYSES

**The drawback of rule-based methods.** Existing reinforcement learning approaches for temporal video grounding are typically rule-based, generating a reasoning process before producing the final prediction. However, in many cases, the reasoning process is not clearly aligned with the final result. As illustrated in the Figure 7, (a) The reasoning contains clear factual errors. For example, it states "They are not laughing at the doorway", while the people in the video are clearly laughing, and the final prediction significantly deviates from the ground-truth IoU; (b) In another case, the reasoning explicitly mentions that the event occurs at 0s, yet the final prediction indicates the event starts at 19.67s. As shown in the Figure 8, the reasoning process of Time-R1 merely describes the overall video without focusing on the specific event of interest, namely the person who is sitting standing up. In contrast, our method focuses on the event itself during the first reasoning step and then adjusts the end time in the second reasoning step.

(a)   Query: person laughing at the doorway

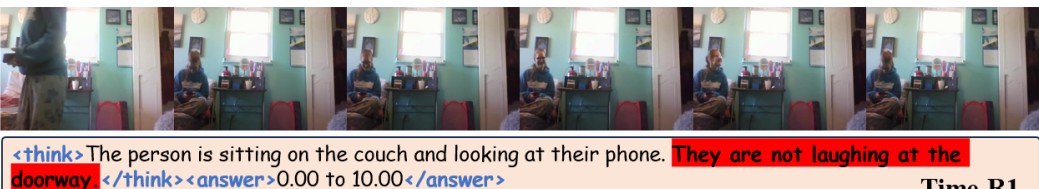

<think>The person is sitting on the couch and looking at their phone. They are not laughing at the doorway.</think><answer>0.00 to 10.00</answer>
                                                                      **Time-R1**

<think>The event "person laughing at the doorway" seems to occur when the person enters the room and continues for a period after they sit down. <timestamp> 2.00 to 12.94 </timestamp>
Considering the event continues beyond the point where the person sits down, the event ends slightly later than the person's laughter stops.<timestamp>1.50 to 13.44</timestamp>
</think><answer>1.50 to 13.44</answer>
                                                                      **TAR-TVG(Ours)**

(b)   Query: person opening a cabinet in a kitchen.

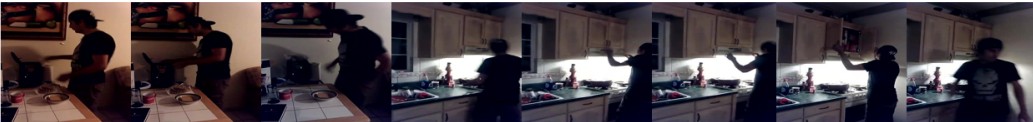

<think>The person is seen opening a cabinet in the kitchen at around 0.00 seconds. The event continues until the person closes the cabinet door.</think>
<answer>19.67 to 28.12</answer>
                                                                      **Time-R1**

<think>The event "person opening a cabinet in a kitchen" seems to occur after the person has walked from the dining area to the kitchen. The context suggests that the person is looking for something specific within the kitchen. <timestamp> 19.40 to 28.04 </timestamp>
Considering the context and the person's actions, the event continues beyond the initial opening of the cabinet. The person appears to be searching through the contents of the cabinet, indicating the event ends once the search is complete <timestamp> 19.00 to 26.44 </timestamp></think> <answer> 19.00 to 26.44 </answer>
                                                                      **TAR-TVG(Ours)**

Figure 7: Illustration of incorrect reasoning produced by the rule-based model VS our TAR-TVG method, showing that our method yields more faithful and correct reasoning traces.

**Test-time self-refinement vs ours.** We compare our approach with a test-time self-refinement strategy, where the model iteratively refines its outputs by conditioning on previous predictions. This differs from conventional test-time augmentation, as it operates on the model's internal reasoning rather than modifying the input modality. However, this strategy does not improve the model's capability in practice that regardless of the number of refinement steps, the predictions remain largely unchanged, and overly long inputs may even degrade performance. As shown in Table 5, increasing the number of refinement steps at test time does not improve performance, in fact, it may lead to a declining trend. For instance, metrics such as R1@0.3 do not show any improvement and can even decrease as the number of refinement iterations grows.

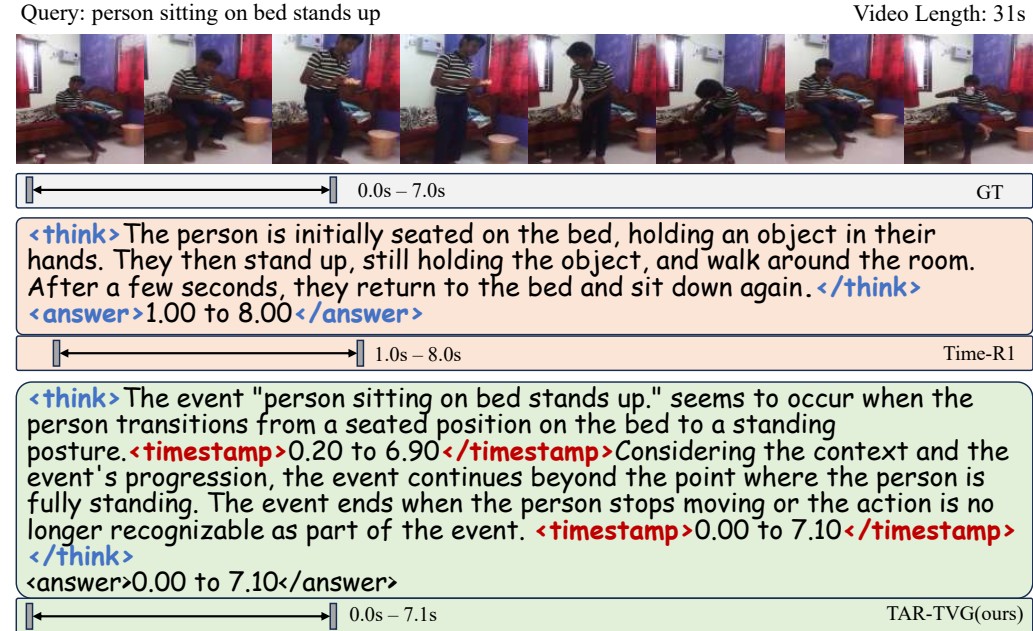

Query: person sitting on bed stands up                                    Video Length: 31s

0.0s − 7.0s                                                                GT

<think>The person is initially seated on the bed, holding an object in their hands. They then stand up, still holding the object, and walk around the room. After a few seconds, they return to the bed and sit down again.</think>
<answer>1.00 to 8.00</answer>

1.0s − 8.0s                                                                Time-R1

<think>The event "person sitting on bed stands up." seems to occur when the person transitions from a seated position on the bed to a standing posture.<timestamp>0.20 to 6.90</timestamp>Considering the context and the event's progression, the event continues beyond the point where the person is fully standing. The event ends when the person stops moving or the action is no longer recognizable as part of the event. <timestamp>0.00 to 7.10</timestamp></think>
<answer>0.00 to 7.10</answer>

0.0s − 7.1s                                                                TAR-TVG(ours)

Figure 8: Examples of thinking processes that are not focused on specific events

In contrast, our method can be regarded as a training-time refinement strategy: by inserting timestamp anchors during training, the model is explicitly guided to progressively produce more accurate temporal spans. As shown in Table 5, applying the test-time self-refinement strategy actually results in performance degradation compared to the baseline, for example, R1@0.7 drops by 1.8%. In comparison, our training-time refinement provides stronger supervision and effectively enhances the model's capability, leading to a 2.7% improvement in R1@0.7 over the baseline.

Table 5: Comparison between our method and the test-time self-refinement strategy.

| Method | mIoU | R1@0.3 | R1@0.5 | R1@0.7 |
|---|---|---|---|---|
| Baseline | 59.8 | 82.4 | 70.6 | 47.5 |
| Test-time refinement(2) | 58.9 | 81.7 | 70.1 | 45.7 |
| Test-time refinement(3) | 58.1 | 80.6 | 70.3 | 45.1 |
| Test-time refinement(4) | 58.7 | 80.2 | 69.9 | 44.8 |
| Ours | **61.1** | **83.6** | **71.4** | **50.2** |

**Think length.** As shown in Table 6, our model achieves better performance by increasing the inference length and strategically inserting timestamp tags. By allowing the model to reason over longer temporal contexts, it can capture more detailed video dynamics and align textual queries more accurately with the corresponding video segments. The inclusion of timestamp tags further guides the model to generate temporally precise predictions, which, together with extended reasoning, leads to the observed improvements across different model sizes.

Table 6: Comparison of GRPO, SFT+GRPO across different model sizes.

| Model | Model Size | GRPO | SFT+GRPO |
|---|---|---|---|
| time-R1 | 3B | 88.7 | 68.4 |
| time-R1 | 7B | 84.5 | 62.6 |
| TAR-TVG | 3B | 186.1 | 185.6 |
| TAR-TVG | 7B | 175.0 | 158.4 |

## B REPRODUCIBILITY AND IMPLEMENTATION DETAILS

In this section, we provide further implementation details to ensure the reproducibility of our work. Specifically, we describe the training parameters, hyperparameter configurations, and the prompts used in our experiments.

### B.1 BASELINE IMPLEMENTATIONS AND ASSOCIATED DATASETS FOR TVG.

- **TimeChat** (Ren et al., 2024) is built upon the InstructBLIP architecture and introduces a video Q-former to encode video tokens. It operates at a resolution of 224 and samples 96 frames. TimeChat is trained on multiple datasets, including Charades-STA (Gao et al., 2017), ActivityNet-Caption (Caba Heilbron et al., 2015), and DiDeMo (Anne Hendricks et al., 2017) et al., as well as additional datasets for video captioning and video summarization.

- **TRACE** (Guo et al., 2024) treats each combination of timestamp, saliency score, and caption as a discrete event, enabling the LVLM to autoregressively generate event sequences. It operates at a resolution of 336 and samples 128 frames. TRACE is trained on dozens of datasets, including ShareGPT4Video (Chen et al., 2024), ActivityNet-Captions (Caba Heilbron et al., 2015), InternVid (Wang et al., 2023), Next-QA (Xiao et al., 2021), and others.

- **TimeSuite** (Zeng et al., 2025) introduces a token shuffling strategy to compress long video token sequences and incorporates positional encoding to enhance visual understanding. It adopts a resolution of 224 and samples 128 frames. TimeSuite is trained on a dataset called TimePro, which is constructed by combining DiDeMo (Anne Hendricks et al., 2017), QuerYD (Oncescu et al., 2021), YT-Temporal (Yang et al., 2023), InternVid (Wang et al., 2023), HowTo100M (Miech et al., 2019), and other datasets.

- **Time-R1** (Wang et al., 2025b) is a reasoning-enhanced post-training framework based on reinforcement learning with verifiable rewards. The LVLM first generates chain-of-thought descriptions and then predicts timestamps. Its post-training is optimized using Generalized Reinforcement Policy Optimization (GRPO) with a reward function that integrates a structured template reward and a timestamp-aware tIoU reward. Time-R1 is trained on a total of 2.5k high-quality samples from the training sets of several datasets, including DiDeMo (Anne Hendricks et al., 2017), QuerYD (Oncescu et al., 2021), YT-Temporal (Yang et al., 2023), InternVid (Wang et al., 2023), and HowTo100M (Miech et al., 2019).

### B.2 TRAINING AND EVALUATION DETAILS

For SFT, we adopt two approaches: full-parameter fine-tuning and LoRA fine-tuning as described in the main text. Since full-parameter fine-tuning in SFT can lead to catastrophic forgetting, we ultimately choose LoRA fine-tuning.

For RL training, we adopt GRPO with full-parameter tuning and generate $G = 8$ reasoning trajectories per sample. We use a learning rate of $9 \times 10^{-7}$ with the AdamW optimizer ($\beta_1 = 0.9$, $\beta_2 = 0.999$), and apply a linear scheduler to decay the learning rate from $9 \times 10^{-7}$ to 0. In the optimization objective, the KL-divergence penalty scaling coefficient $\alpha$ is set to 0.005.

For the 3B model, SFT Traing takes approximately 13 hours(1epoch batchsize 4) on 4×NVIDIA A100 40GB GPUs. RL Training takes approximately 40 hours(batch size 8, 2epoch) on 8×NVIDIA A100 40GB GPUs. For the 7B model, SFT Traing takes approximately 15 hours(1epoch batchsize 4) on 4×NVIDIA A100 40GB GPUs. RL training takes about 60 hours(batch size 16 ,3epoch) on 16×NVIDIA A100 40GB GPUs.

To balance training cost and performance, we adopt a frame sampling rate of 2 fps for training on Charades-STA, resulting in an average of approximately 70 frames per video. For QVHighlights, we use a lower sampling rate of 0.5 fps, yielding an average of around 75 frames per video. Additionally, each frame is resized to maintain a total pixel count of roughly 80,000.

When evaluated on ActivityNet-Caption, Charades-STA, QVHighlights, and TVGBench, all videos are uniformly sampled at 2 FPS, and the total number of pixels per video is 2.8 million.

### B.3 ALGORITHM DETAIL

Algorithm 1 shows that Python implementation of the Temporal Anchor-Constrained Reward (TAR). Computes the reward for predicted temporal anchors based on weighted sIoU ($\text{TAR}_{sIoU}$), anchor count penalty ($\text{TAR}_{num}$), and progressive refinement ($\text{TAR}_{refine}$).

---

**Algorithm 1** Python snippet of TAR Reward

---

Initialize related package

```
def tar_reward(sIoUs, s, beta=5.0, gamma=1.0):
    import numpy as np
    sIoUs = np.array(sIoUs)
    hat_s = len(sIoUs)
    tar_sIoU = sum((i + 1) * sIoUs[i] for i in range(hat_s))
    delta = [1 if i == 0 or sIoUs[i] > sIoUs[i-1] else -1 for i in range(hat_s)]
    tar_refine = sum(delta[1:])
    tar_num = (hat_s - s) ** 2
    r_tar = tar_sIoU + beta * tar_refine - gamma * tar_num
    return r_tar
```

---

### B.4 PROMPT DETAIL

We use the following prompt (see Figure 9). Our prompt initially guides the model to generate outputs in the format of `<think></think>` `<timestamp></timestamp>` `<think></think>` `<answer></answer>`. To align with the mainstream format of `<think></think><answer></answer>`, we further convert it into the format of `<think>` `<timestamp></timestamp>` `<timestamp></timestamp>` `</think>` `<answer></answer>`.

To accurately pinpoint the event "[EVENT]" in the video, please provide your reasoning process in two separate <think> blocks.

In the first <think> block, describe your initial understanding of when the event occurs, including some key time points or time intervals related to the event to support your reasoning (e.g. event starts around 12.5 seconds, key moment at 14.8 seconds, important interval between 15.0 and 16.5 seconds). Use natural language to explain the context and timing.

Then, provide your first estimated time range using a <timestamp> tag in the format "start time to end time" (e.g., <timestamp>12.54 to 17.83</timestamp>).

In the second <think> block, continue by reflecting on the evidence, analyzing the event with respect to its temporal context and any additional time points or intervals that help clarify or refine your understanding. Elaborate on the event's progression or related contextual cues that impact your estimation.

Finally, output your confirmed result on a new line using the format <answer>start time to end time</answer>.

Figure 9: prompt

To address the issue in previous methods where the prompt allowed the model to freely output as long as the final result was a correct temporal location, we improve our prompt design. Before generating the first `<timestamp>`, the model is guided to provide an initial understanding of the event, incorporating the relevant context and key temporal points. Before generating the second `<timestamp>`, the model is instructed to revisit the previous thought process, reanalyze the video, verify the correctness of the first timestamp, and revise or adjust it if necessary. With this improved prompting strategy, the model can better comprehend the video content and make use of both textual and visual contexts to achieve more accurate temporal localization.

## C ABLATION STUDY

**Ablation study on the number of generated answers during GRPO training.** As shown in the results, increasing the number of generated candidates $G$ in GRPO leads to consistent improvements in performance. When $G$ increases from 2 to 8, the mean Intersection over Union (mIoU) improves from 58.8 to 59.3, and the recall at higher IoU thresholds also shows steady gains, with R1@0.7

increasing from 44.9 to 45.9. These trends suggest that generating more samples helps the model better capture the temporal boundaries of the target segments. However, considering the limitations in computational resources and generation time, we restrict the maximum number of generated candidates to $G = 8$ in our experiments.

Table 7: Results with different numbers of generated answers (G) during GRPO training

| G | mIoU | R1@0.3 | R1@0.5 | R1@0.7 |
|---|------|--------|--------|--------|
| 2 | 58.8 | 81.8 | 69.3 | 44.9 |
| 4 | 59.0 | 81.7 | 69.4 | 45.1 |
| 8 | 59.3 | 81.5 | 70.1 | 45.9 |

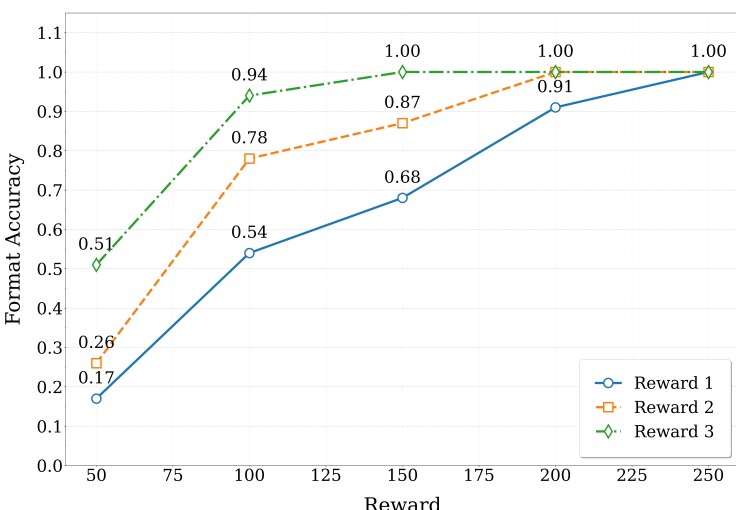

Figure 10: Format reward.

**Ablation on Format reward.** We investigate how different reward magnitudes assigned to the format token affect the learning efficiency of the model. As shown in the Figure 10, when the format reward is set to a higher value (e.g., 3), the model is able to learn the correct output format much more quickly. Specifically, with a reward of 3, the normalized format accuracy reaches 0.94 at just 100 steps and achieves a perfect score of 1 by step 150. In contrast, smaller reward values such as 1 or 2 lead to slower improvements, requiring up to 200 or even 250 steps to reach comparable performance. These results suggest that assigning a higher reward to correct format generation provides a strong learning signal, enabling the model to acquire format correctness more efficiently in the early stages of training. Since the model has basically learned the format within 150 steps when the reward is set to 3, we set the maximum reward to 3.

Table 8: Ablation results for different values of $\beta$ and $\gamma$.

| $\beta$ | $\gamma$ | mIoU | R1@0.5 | R1@0.7 |
|---------|----------|------|--------|--------|
| 1 | 5 | **59.3** | **70.1** | **45.9** |
| 2 | 5 | 58.1 | 68.5 | 44.2 |
| 3 | 5 | 57.1 | 67.9 | 44.3 |
| 1 | 4 | 45.7 | 58.1 | 37.5 |
| 1 | 3 | 43.2 | 57.9 | 35.1 |
| 1 | 2 | 39.1 | 49.8 | 32.4 |

**Ablation on $\beta$ and $\gamma$.** We conducted a simple ablation study on $\beta$ and $\gamma$ in Table 8. First, we fixed $\gamma$ at 5, which serves as a penalty term to encourage the model to generate the correct number of timestamps. As we gradually increased $\beta$, the localization performance slightly decreased (mIoU

dropped from 59.3 to 58.1). This is because the model tends to gain larger rewards from refinement rather than improving the IoU, resulting in smaller overall rewards.

Next, we fixed $\beta$ at 1 and gradually decreased $\gamma$. In this case, the model's localization ability dropped significantly (mIoU as low as 39.1), since the penalty on generating an incorrect number of timestamps was weakened, leading the model to exploit the reward function improperly.

## D    QUALITATIVE RESULT

**Case study of short video on Charades-STA.** Figure 11 demonstrates our model's understanding and localization capabilities on short videos. Compared to other models, our model accurately identifies the exact start and end times of the action. In contrast, other models either predict an incorrect start time for the "throws" action or include unrelated actions within the predicted segment, failing to precisely detect the beginning and end of the throwing action.

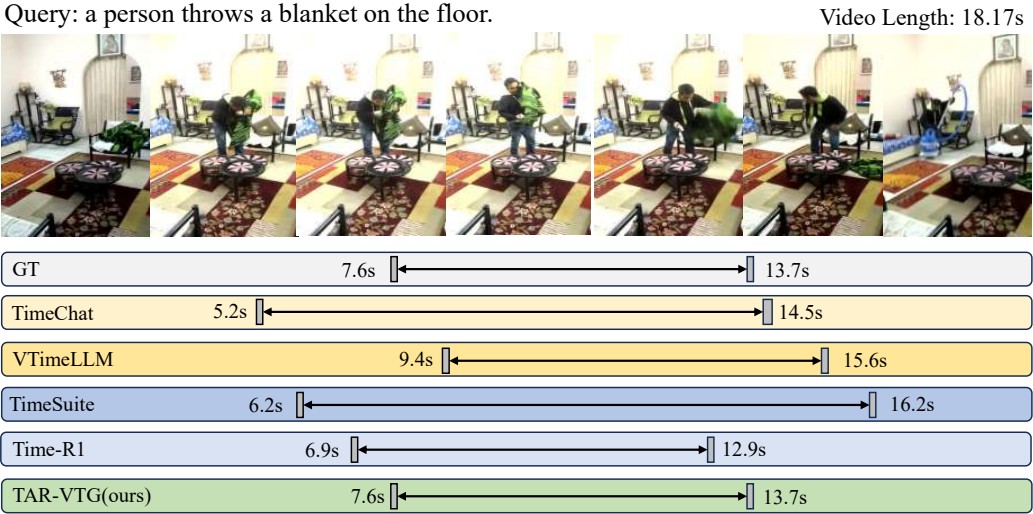

Query: a person throws a blanket on the floor.                    Video Length: 18.17s

**Ours timestamp anchor-based think:**

<think>The event "a person throws a blanket on the floor" seems to occur after the person picks up the blanket from the chair and walks towards the floor. <timestamp>7.90 to 13.43</timestamp>Considering the person's movement and the context, the event continues beyond the point where the blanket is on the floor. The person appears to be fully engaging with the blanket, indicating the event ends once the action is complete. <timestamp>7.60 to 13.73</timestamp> </think>

Figure 11: our success case 1

**Case study of long video on ActivityNet-Caption.** Figure 12 shows the localization performance of our method on a longer video. Our model accurately identifies the segments where the woman approaches the stage and the man sits on the chair, as described in the query. In contrast, other models include either unrelated actions or miss the part where the woman approaches the stage.

## E    LIMITATIONS

Although our proposed TAR-TVG method demonstrates promising performance, it still has several limitations. First, the method is specifically designed for Temporal Video Grounding (TVG) and may not generalize well to other video understanding tasks. Second, due to the nature of the training strategy, the method suffers from relatively slow training and inference speeds. Lastly, constrained by GPU resources, we perform frame sampling and spatial downsampling on individual frames, which may lead to a degradation in temporal modeling performance. Figure 13 shows a failure case of our method. The model outputs a timestamp in the first step that significantly deviates from the correct IoU range, and in the second step, it fails to fully correct the initial timestamp. As a result, the final prediction yields a suboptimal outcome.

Query: A young woman walks onto a stage and speaks to a man sitting in a chair.  Video Length 236.49s

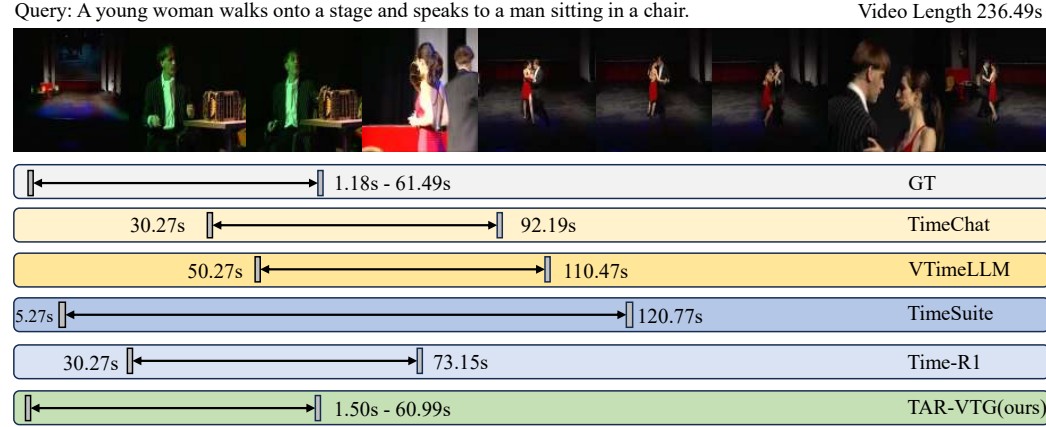

| | |
|---|---|
| 1.18s - 61.49s | GT |
| 30.27s — 92.19s | TimeChat |
| 50.27s — 110.47s | VTimeLLM |
| 5.27s — 120.77s | TimeSuite |
| 30.27s — 73.15s | Time-R1 |
| 1.50s - 60.99s | TAR-VTG(ours) |

**Ours timestamp anchor-based think:**

<think>The event "A young woman walks onto a stage and speaks to a man sitting in a chair." seems to occur from the moment the woman enters the frame until she stops speaking and the man responds.<timestamp>2.09 to 60.43</timestamp> Considering the context and the event's progression, the event continues beyond the point where the woman stops speaking and the man begins to respond. The event ends when the woman leaves the stage.<timestamp>1.50 to 60.99 </timestamp> </think><answer>1.50 to 60.99</answer>

Figure 12: our success case 2

Query: person eating a sandwich.

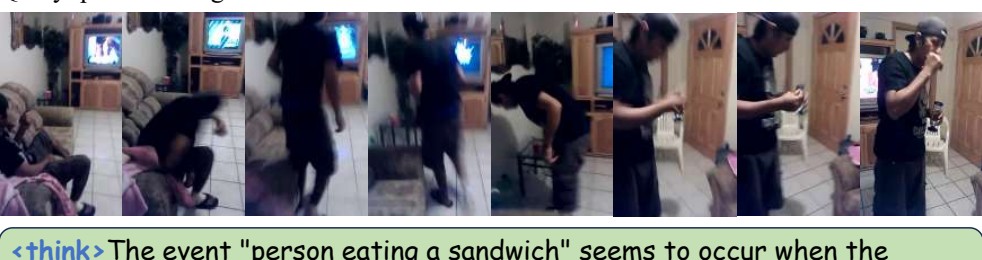

<think>The event "person eating a sandwich" seems to occur when the person is visibly holding and interacting with an object resembling a sandwich. The context suggests the person is seated comfortably on the couch while watching television.
<timestamp>0.10 to 19.63</timestamp>
Considering the context and the movement of the person, the event continues beyond the initial observation point. The person's posture and hand movements indicate continuous engagement with the food item.
<timestamp>0.00 to 19.73</timestamp>
</think>
<answer>0.00 to 19.73</answer>

| | | | |
|---|---|---|---|
| GT | 34.90s – 45.00s | | |
| Timestamp 1 | 0.10s – 19.63s | ☹ | |
| Timestamp 2 | 0.00s – 19.73s | ☹ | |

Figure 13: our failure case

# F USE OF LARGE LANGUAGE MODELS

We used large language models (LLMs) in a limited and auxiliary manner to aid the preparation of this paper. Specifically, LLMs were employed to polish grammar and improve readability of the manuscript.

