# OpenReview forum: "TAR-TVG: Enhancing LVLMs with Timestamp Anchor-Constrained Reasoning for Temporal Video Grounding"
_ICLR.cc/2026/Conference — Submitted to ICLR 2026_

### Official Review · Reviewer_EVwv · 2025-10-24

**Soundness:** 2
**Presentation:** 2
**Contribution:** 2
**Rating:** 4
**Confidence:** 3

**Summary:**

This paper propose TAR-TVG, a novel timestamp anchor-constrained reasoning framework for temporal video grounding, which includes a efficient reinforcement learning strategy for extracting high-quality reasoning traces. The experiments reveal the improved performance for temporal video grounding with verifiable reasoning chains for progressively refined temporal estimations.

**Strengths:**

1. The proposed method adopts a three-stage training process with reinforcement learning, improving the interpretability and accuracy of temporal video grounding.

2. The experiment result is solid with high performance on several evaluation benchmarks for temporal video grounding.

3. Convincing visualization examples are provided to prove the effectiveness of the proposed method.

**Weaknesses:**

1. Only testing on temporal video grounding tasks would be limited for the proposed method adopted with LVLMs. Many temporal-aware LVLMs also demonstrate effective generalization ability for related video understanding tasks, not limited to temporal video grounding only. I encourage the authors to evaluate the proposed method on more temporally related video understanding benchmarks.

2. The challenges claimed by this paper, that ‘ the prompts of the used method only implicitly guide the model to output timestamp tags, often leading to missing, incorrect-formatted, or irrelevant tags’, are relatively weak. Since quite a few LVLM-free methods, such as FlashVTG, achieve good temporal video grounding performance and do not have such challenges. The authors should reorganize the statement of addressed challenges.

**Questions:**

1. See weakness.

2. Why adopt a three-stage training process in the order of RL-SFT-RL instead of processes(such as SFT-RL only) in other orders? This may need to be better clarified by more ablation experiments with both evaluation performance and training cost.

3. The paper may require a small adjustment of compilation format, such as citation font color in the main text and the underline in the reference, which is different from papers from previous years and other reviewed papers, and may be caused by the compilation.

---

> ### Author Response · Authors · 2025-11-21
>
> We sincerely appreciate the time and expertise you have devoted to reviewing our submission. Our responses are provided below.
>
> **[W1] Test on VQA task**
>
> To evaluate our method on VQA tasks, we tested TAR-TVG on two comprehensive video understanding benchmarks (MVBench and VideoMME) in a zero-shot setting. As shown in the table below, although TAR-TVG is trained only on TVG tasks, it outperforms the Qwen2.5-VL-7B baseline on VQA.
> |         Model      | MVbench | VideoMME |
> |---------|---------|----------|
> | Qwen2.5-VL-7B | 51.5    | 48.8     |
> | Ours          | 52.8    | 52.9     |
>
> **[W2] Challenges of LVLM-Based Approaches Are Not Those of FlashVTG**
>
> FlashVTG represents a class of traditional methods that rely on pre-extracted offline features and task-specific modules, whereas our work focuses on the end-to-end reasoning capabilities of LVLMs. These are two distinct research directions, facing different challenges. We will revise our wording to avoid any potential confusion.
>
> **[Q2] Clarification on the "RL-SFT-RL" Pipeline**
>
> While the process involves three steps, it is designed to address the "cold-start" problem inherent in reasoning tasks. Our method follows the standard SFT-RL paradigm, where the first RL stage is used solely for data collection.
>
> **Stage 1 (Data Synthesis)**: This is an automated data mining process by training model to collect data. Since the base model (Qwen2.5-VL-7B) initially struggles to follow the strict timestamp format required for rewards (as detailed in Section 3.5), direct RL optimization is inefficient. We train the base model to explore and filter for high-quality reasoning traces without needing a larger model. The model from this stage is not used for subsequent training."
>
> **Stage 2 (SFT)**: We use this self-generated data (30k samples) to "warm up" the original model (Qwen2.5-VL-7B), ensuring it adheres to the output format and reasoning logic.
>
> **Stage 3 (RL)**: This is the actual optimization phase using GRPO.
>
>
> **[Q3] Adjustment of Compilation Format**
>
> We thank the reviewer for the detailed observation regarding the compilation format. Regarding the citation font color, we adopted the style used in recent ICLR publications, such as TimeSuite (ICLR 2025), to maintain consistency with contemporary formatting trends. However, we fully agree with your suggestion regarding the underlines in the references, and we will remove them in the revised manuscript to ensure a cleaner and more standard presentation.

---

### Official Review · Reviewer_jaMU · 2025-11-01

**Soundness:** 3
**Presentation:** 2
**Contribution:** 2
**Rating:** 4
**Confidence:** 5

**Summary:**

The paper proposes Timestamp Anchor-constrained Reasoning (TAR-TVG) for temporal video grounding with Large Vision-Language Models (LVLMs). Instead of merely prompting a model to add <timestamp> tags inside chain-of-thought, TAR-TVG explicitly constrains the reasoning with (1) a format reward that enforces valid <think>…</think>, <timestamp>…</timestamp>, and <answer>…</answer> structures, (2) a soft IoU reward that can be negative to provide graded feedback even with non-overlapping segments, and (3) a timestamp anchor reward that weights later anchors more and rewards progressive refinement of timestamps. Training follows a three-stage RL→SFT→RL routine: initial GRPO to mine ~30k high-quality CoT traces, SFT on those traces, and final GRPO with anchor constraints. On Charades-STA and QVHighlights, the method reports state-of-the-art or competitive results (e.g., mIoU 61.1 and R1@0.7 50.2 on Charades-STA with a 7B LVLM), and shows zero-shot gains on ActivityNet-Captions and TVGBench.

**Strengths:**

1. The motivation is clear:
Identifies a concrete failure mode of prior “prompt-only” reasoning and answers it with explicit, verifiable anchors coupled to the final answer. The progressive-refinement constraint is especially compelling and differentiates TAR-TVG from previous works which supervise format or outcomes but not intermediate time anchors directly.

2. Three-stage RL→SFT→RL training is a effective way of collecting SFT data.
Mining 30k CoT traces with explicit anchor quality thresholds, then SFT, then RL again is an effective pipeline that improves the rate of valid-format reasoning and final accuracy. The paper quantifies each stage’s contribution.

3. The ablation study is comprehensive.

**Weaknesses:**

1. presentation can be improved:
The text in figure 1 and 3 is too small and hard to see. Please make them bigger.

The well-known background such as GRPO and some trivial implementation such as format reward can be moved to appendix. Make more room for your ablation study which is more interesting.

2. the contribution is limited:
The main innovations are (1) explicitly include <timestamp></timestamp> in the reasoning process. This is mainly about output formating.
(2) reward to encourage predicting progressively improved time periods. An ad-hoc design of reward functions which is mainly based on empirical observation.
(3) leveraging some criteria and a pretrained model to produce SFT data. This method has been widely adopted in many previous papers [1].

3. Lack comparison to previous RL based method such as Video-R1 [1]
[1] Video-R1: Reinforcing Video Reasoning in MLLMs

**Questions:**

1. In the first stage of the RL-SFT-RL TRAINING STRATEGY, do you optimize the model or not? If you only collect data in this stage without training the model, please remove the term GRPO since it means you optimize the model.

---

> ### Author Response · Authors · 2025-11-21
>
> Thank you for your thoughtful review. We sincerely appreciate your time and constructive feedback. Our responses are provided below.
>
> **[W1] Presentation**
>
> We thank the reviewer for the valuable suggestions regarding the presentation. We will redesign Figure 1 and Figure 3 with larger fonts to improve readability. We will move the standard background GRPO to appendix.
>
> **[W2] Contribution**
>
> We thank the reviewer for the insightful comments and offer the following clarifications:
>
> 1.  The use of explicit `<timestamp></timestamp>` tags is a key mechanism for improving **interpretability**. By inserting verifiable temporal anchors and refining them via **RL**, the model’s reasoning becomes more faithful and correct.
> 2.  Our **progressive  reward design** is grounded in the human “coarse-to-fine” reasoning pattern. The $TAR_{\text{refine}}$ reward enforces that each new temporal anchor must be more accurate than the previous one, and its effectiveness is confirmed by our ablation results (**Table 4**), where removing it reduces **R1@0.7** from **50.2** to **47.93**.
> 3.  While **SFT data generation** using pretrained models is common, our approach differs from distilling large foundation models (e.g., **GPT-4o**). We use **RL** to let a relatively small **7B** model self-generate and refine data, then apply strict filtering, providing a more **economical and scalable pipeline**.
>
> We hope these clarifications accurately convey our contributions.
>
> **[W3] Clarification on Comparison with Video-R1**
>
> We thank the reviewer for bringing up Video-R1. We have already cited and discussed this work in Section 2. However, since Video-R1 does not report results on temporal video grounding, a direct quantitative comparison is unfortunately not available.
>
> **[Q] Clarification on Model Optimization in Stage 1**
>
> We appreciate the reviewer's suggestion and confirm that we do optimize the model parameters using GRPO during the first stage.  This optimization is crucial as it enables the model to discover and produce increasingly high-quality reasoning traces that a static base model would fail to generate. We collect and filter these self-generated samples throughout the optimization trajectory to curate the final 30k dataset. We emphasize that while the model is trained to facilitate this data synthesis, the resulting model weights are discarded, and only the curated data is retained for the subsequent SFT stage.

---

### Official Review · Reviewer_PYVz · 2025-11-01

**Soundness:** 3
**Presentation:** 3
**Contribution:** 2
**Rating:** 4
**Confidence:** 4

**Summary:**

This paper proposes TAR-TVG, a reinforcement learning method for Temporal Video Grounding (TVG) that introduces explicit constraints on timestamp anchors within the model's reasoning process. The core innovation involves designing reward functions that enforce: (1) correct formatting of timestamp tags, (2) progressive refinement of temporal predictions (later timestamps must be more accurate than earlier ones), and (3) control over the number of generated timestamps. To address training instability, the authors employ a three-stage RL→SFT→RL strategy that automatically generates high-quality Chain-of-Thought data. The method achieves state-of-the-art results on Charades-STA and shows strong performance on QVHighlights, ActivityNet-Captions, and TVGBench.

**Strengths:**

1. The introduction of timestamp anchors and the progressive refinement reward (TARrefine) is a conceptually novel and well-motivated approach. It explicitly encourages the model to mimic a human-like, coarse-to-fine reasoning process, which is a clear advancement over prior RL-based methods that only implicitly prompted for timestamps.

2. The paper demonstrates compelling state-of-the-art performance on multiple established benchmarks (Charades-STA, QVHighlights). The improvements over strong baselines like Time-R1 are significant and well-documented across various metrics (mIoU, R1@0.5, R1@0.7).

3. The proposed RL→SFT→RL pipeline is a pragmatic solution to the cold-start problem where base models fail to generate initial timestamp tags. The method of automatically curating a high-quality CoT dataset from initial RL rollouts is efficient and eliminates the need for manual annotation.

**Weaknesses:**

1. The proposed method is a complex, multi-stage pipeline (RL→SFT→RL) built upon another complex framework (GRPO). This "pipeline-ception" raises concerns about reproducibility, computational cost, and practicality in general. The need for such a heavy-handed approach suggests that the core idea might be fragile or complex, making it challenging to optimize directly.

2. The three-stage training process is exceptionally resource-intensive. The first RL stage is acknowledged to have a low success rate for generating sound samples, making it highly inefficient. Training requires up to 16 A100 GPUs for 60+ hours. This level of resource consumption poses a significant barrier for most researchers, limiting the practical adoption and verifiability of the work.

3. The method is highly specialized for the Temporal Video Grounding task. The heavy reliance on specific output formatting (<think>, <timestamp>) and custom rewards makes it non-trivial to adapt to other video reasoning tasks (e.g., captioning, VQA). The paper does not demonstrate the generality of the "progressive anchor" concept beyond TVG.

4. While the method produces "reasoning chains," the evaluation is solely based on the final grounding accuracy (IoU). There is no qualitative or quantitative analysis of the faithfulness or correctness of the generated reasoning itself. The examples in Appendix A highlight that previous models produce flawed logic; however, it remains unproven whether TAR-TVG's reasoning is truly more logical or faithful, or if it has simply learned to exploit the reward structure by placing correct timestamps within a templated text.

5. The method's success is closely tied to a carefully engineered prompt (detailed in Appendix B.4) that explicitly guides the two-step reasoning process. The performance gains might be partially attributable to this superior prompt design rather than the RL reward mechanism alone. An ablation where the same prompt is given to a strong baseline is missing.

**Questions:**

While TAR-TVG presents a novel idea and achieves strong results, the combination of extreme complexity, high computational cost, lack of demonstrated generalization, and unresolved questions about the true nature of the learned reasoning leads me to lean towards a weak rejection. The core idea of progressive anchor refinement is promising, but the current execution feels overly engineered and inefficient for the gains achieved. I would be willing to reconsider my decision if the authors can convincingly address the concerns above, particularly those related to generality and cost-effectiveness.

---

> ### Author Response · Authors · 2025-11-21
>
> We would like to thank the insightful comments and valuable feedback.
>
> (Due to the length of the response exceeding the limit, we will reply in two parts. This is the first part.)
>
> **[W1] Complexity Analysis**
>
> **Clarification on the "Three-Stage" Pipeline**: While the process involves three steps, it is designed to solve the "cold-start" problem inherent in reasoning tasks, similar to the method used in DeepSeek-R1.
>
> **Stage 1 (Data Synthesis)**: This is an automated data mining process by training model to collect data. Since the base model (Qwen2.5-VL-7B) initially struggles to follow the strict timestamp format required for rewards (as detailed in Section 3.5), direct RL optimization is inefficient. We train the base 7B model to explore and filter for high-quality reasoning traces without needing a larger model (e.g. GPT-4o) to collecting data. **The model from this stage is not used for subsequent training**.
>
> **Stage 2 (SFT)**: We use this self-generated data (30k samples) to "warm up" the original model, ensuring it adheres to the output format and reasoning logic.
>
> **Stage 3 (RL)**: This is the actual optimization phase using GRPO.
>
> In summary, our method is relatively lightweight, incorporating a few verifiable anchor in reason trace and optimizing model through data collection combined with a standard SFT-RL pipeline.
>
> **[W2] Efficiency analysis**
>
> We thank the reviewer for examining the computational resources required. We would like to clarify that the reported training duration is largely an artifact of our specific hardware limitations rather than an inherent inefficiency of the method.
>
> **Hardware Bottlenecks.** Our experiments were conducted using 16× NVIDIA A100 PCIe 40GB GPUs. Crucially, these gpus lack NVLink support and have limited VRAM. To train the model, we were forced to employ DeepSpeed ZeRO-3 Offload, which introduces significant communication overhead between the CPU and GPU. The limited bandwidth of the PCIe architecture exacerbated this bottleneck, severely slowing down the training process.
>
> **If we used GPUs equipped with NVLink, our training time would be reduced by half or even more.**
>
> **[W3] Generality to other tasks**
>
> **Zero-Shot Generalization on VQA.** To verify whether our specialized training compromises general VQA capabilities, we evaluated TAR-TVG on two comprehensive video understanding benchmarks—MVBench and VideoMME—in a zero-shot setting. Importantly, all results were obtained using standard direct generation (without activating the task-specific reasoning/thinking procedure), ensuring a fair comparison of the models’ intrinsic representation abilities. As shown in the table below, TAR-TVG not only avoids catastrophic forgetting but even surpasses the Qwen2.5-VL-7B baseline, since TVG is fundamentally a subtask of video understanding. Effective temporal grounding is crucial for  VQA because it helps the model focus on the most relevant moments while reducing distractions from irrelevant or redundant content. For instance, on VideoMME, our method achieves a 4.1 improvement over the base model.
>
> |         Model      | MVbench | VideoMME |
> |---------|---------|----------|
> | Qwen2.5-VL-7B | 51.5    | 48.8     |
> | Ours          | 52.8    | 52.9     |
>
> **Potential for Method Generalization.** While our current implementation uses timestamp-style anchors designed for **TVG task**, the core idea of **Anchor-Constrained Reasoning** can also be applied to **VQA tasks**. In this setting, anchors can be redefined as intermediate logical steps. For example, a reasoning chain could iteratively examine and update candidate answers (e.g., `<think> analysis → <anchor>Option A</anchor> → re-evaluation → <anchor>Option B</anchor> </think>`), enabling the model to explicitly revise its initial guess. Developing verifiable reward $R_{\text{acc}}$ to verify answer correctness, and refinement reward $R_{\text{refine}}$ to reward revised answers, offers a promising direction for extending our method beyond temporal reasoning.

---

> ### Author Response · Authors · 2025-11-21
>
> (Due to the length of the response exceeding the limit, we will reply in two parts. This is the second part.)
>
> **[W4] Faithfulness of TAR-TVG' reasoning**
>
> **Qualitative Analysis.** To verify the faithfulness of our reasoning, we update Figure 7 with a direct comparison against Time-R1. The case study shows that our model aligns more closely with the visual content. For example, for the query “person laughing at the doorway,” Time-R1 incorrectly states that the person is not laughing, while TAR-TVG correctly identifies the event and further refines the end boundary in its second reasoning step. This demonstrates that our reasoning is grounded in real visual evidence rather than generic templates, helping reduce hallucinations.
>
> **Quantitative Analysis.** To verify that our improvements arise from genuinely better reasoning, we conducted a counterfactual substitution experiment. Specifically, we replaced TAR-TVG’s own reasoning before the first predicted timestamp with reasoning traces generated by Time-R1, and then required the model to continue the reasoning. As shown in the table, this substitution results in a clear performance drop (e.g., mIoU decreases from 61.1 to 58.9 and R1@0.7 from 50.2 to 45.7). This demonstrates that TAR-TVG’s reasoning is not superficial: its reasoning steps play a causal and substantive role in improving localization accuracy.
>
> |                                  | R1@0.3 | R1@0.5 | R1@0.7 | mIoU |
> |----------------------------------|--------|--------|--------|------|
> | TAR-TVG (w/ Time-R1's reasoning) | 81.7   | 70.1   | 45.7   | 58.9 |
> | TAR-TVG (Original)               | 83.6   | 71.4   | 50.2   | 61.1 |
>
> **[W5] Ablation on Superior  Prompt Design Applied to Stronger Baselines**
>
> To verify that our performance gains come from **RL reward** rather than simple prompt design, we conducted an ablation study by applying **TAR-TVG**’s specialized prompt to strong baseline models including **Mimo-VL-7B**, **Qwen3-VL-8B**, and **Time-R1**.
>
> |              | R1@0.3 | R1@0.5 | R1@0.7 | mIoU |
> |--------------|--------|--------|--------|------|
> | Mimo-VL-7B   | 32.6   | 15.9   | 5.2    | 21.9 |
> | Qwen3-VL-8B  | 56.9   | 29.7   | 11.5   | 35.4 |
> | Time-R1      | 81.3   | 69.9   | 46.5   | 59.1 |
> | TAR-TVG      | 83.6   | 71.4   | 50.2   | 61.1 |
>
> The results show that prompting alone cannot reproduce our performance (for example, **Qwen3-VL** achieves only **35.4 mIoU**, far below **TAR-TVG**’s **61.1**).
>
> Further analysis reveals that the baselines either:
> 1. Fail to follow the strict `<think> ... <timestamp>` format (e.g., **Mimo-VL**).
> 2. Although following the format, are unable to effectively refine timestamps in the second step (e.g., **Time-R1**, **Qwen3-VL-8B**).
>
> This demonstrates that the prompt merely provides a structural template guideness, while the actual **progressive refinement capability** must be learned through our **RL framework**.
>
>
> **[Q] About Generality and Cost**
>
> As detailed in **[W1]** and **[W2]**, our training cost is consistent with standard RL approaches once hardware bottlenecks are addressed. Furthermore, as shown in **[W3]**, our method can be generalized to VQA tasks. We hope these clarifications help provide a clearer understanding of our approach and its contributions.

---

### Official Review · Reviewer_EJ9d · 2025-11-11

**Soundness:** 3
**Presentation:** 3
**Contribution:** 3
**Rating:** 6
**Confidence:** 4

**Summary:**

This paper presents Timestamp Anchor-Constrained Reasoning for Temporal Video Grounding (TAR-TVG). The proposed method introduces intermediate timestamp anchors during a reasoning chain (in a large vision-language model) and enforces that each reasoning step progressively improves the timestamp prediction. The training is a three-stage process: 1) initial RL (GRPO) to generate high-quality reasoning traces with anchors; 2) supervised fine-tuning (SFT) on that distilled data; and 3) final RL fine-tuning with the anchor constraints. Experiments on standard TVS benchmarks (Charades-STA, QVHighlights) show that the proposed method achieves state-of-the-art performance and improves interpretability via the anchor-based reasoning chains.

**Strengths:**

**[S1]** The paper is well-written and easy to follow.

**[S2]** The proposed method is interesting, and the mechanism is well-designed to enable the model to produce accurate predictions. Especially, the proposed soft IoU reward is simple yet effectively complements the timestamp anchor-constrained reward.

**[S3]** The interpretability angle (reasoning chains with anchors) is a positive addition.

**Weaknesses:**

**[W1]** Efficiency analysis
- The requirement for heavy RL training (30K reasoning traces, etc) may limit reproducibility and practical adoption. Providing full budget, hardware, and training time details would be helpful to highlight the contribution of the proposed method.

**[W2]** Ablation study
- It is not clear how much of the performance gain is from the anchor mechanism vs simply using more RL/training data/backbone size.
- The interpretability angle is a bit weak. The interpretability claim is only meaningful if users inspect the reasoning traces. The paper should include many qualitative examples and maybe user studies.

**[W3]** Minor issue
- If the method uses multiple anchors, but the best ablation turns out to be just two anchors, then the general claim “introduce anchors (e.g. more than three) and progressive refining” may over-claim the breadth of benefit. The paper should avoid implying many‐anchor advantages if two is sufficient.

**Questions:**

Please see the weaknesses.

---

> ### Author Response · Authors · 2025-11-21
>
> We sincerely appreciate the time and effort you have devoted to providing these detailed observations and questions. We have carefully considered your feedback and provide our responses and proposed revisions below. We hope that these clarifications and updates address your concerns and contribute to improving the manuscript.
>
> **[W1] Efficiency analysis**
>
> The 30K dataset will be released publicly, enabling future research to use it directly without repeating the data-collection process. The associated computational cost and hardware resources are summarized below:
>
> 7B Model
> | Stage           | Data  | Epoch | GPU                          | Time |
> |-----------------|-------------|-------|------------------------------|------|
> | Data Collection | 5K TVG Pairs        | 2     | 16× NVIDIA A100 PCIe 40GB    | 40h  |
> | SFT             | 30K  TVG Pairs       | 1     | 4× NVIDIA A100 PCIe 40GB     | 15h  |
> | RL              | 12K  TVG Pairs      | 3     | 16× NVIDIA A100 PCIe 40GB    | 60h  |
>
> 3B Model
> |Stage| Data  | Epoch | GPU                          | Time |
> |-----|-------------|-------|------------------------------|------|
> | SFT |  30K  TVG Pairs         | 1     | 4× NVIDIA A100 PCIe 40GB     | 13h  |
> | RL  | 12K  TVG Pairs         | 2     | 8× NVIDIA A100 PCIe 40GB     | 40h  |
>
> **[W2] More Ablation study**
>
> **1. Performance gain from the anchor mechanism**
>
> | Model Size | Method                 | DataScale                 | R1@0.3   | R1@0.5   | R1@0.7   | mIoU     |
> |------------|------------------------|---------------------------|----------|----------|----------|----------|
> | **3B** | Time-R1* (Baseline)    | Charades, Qvhighlight     | 79.5     | 65.1     | 37.5     | 55.3     |
> |            | **Ours** (Anchor)      | **Charades, Qvhighlight** | **81.0** | **67.2** | **40.6** | **57.0** |
> |            | 🔺 **Gain from Anchor** | *(Same Data)* | *+1.5* | *+2.1* | *+3.1* | *+1.7* |
> | ---        | ---                    | ---                       | ---      | ---      | ---      | ---      |
> | **7B** | Time-R1* (Baseline)    | Charades, Qvhighlight     | 82.4     | 70.6     | 47.5     | 59.8     |
> |            | **Ours** (Anchor)      | **Charades, Qvhighlight** | **83.6** | **71.4** | **50.2** | **61.1** |
> |            | 🔺 **Gain from Anchor** | *(Same Data)* | *+1.2* | *+0.8* | *+2.7* | *+1.3* |
>
> As shown in the table, the comparison between "Ours (Anchor)" and the "Baseline (Time-R1)"* is conducted under the exact same DataScale (Charades, Qvhighlight) and same Model Size (3B/7B). The rows labeled "Gain from Anchor (Same Data)" explicitly quantify the performance improvement derived solely from the proposed anchor mechanism, ruling out the influence of additional data or backbone size.
>
> **2. Additional Interpretable Examples**
>
> We have added more interpretable examples in the appendix, including Figures 7, 8, 11, and 12. Figure 7 illustrates that our reasoning process is more faithful and accurate compared to Time-R1, aligning better with the actual video content. Figure 8 highlights that our reasoning focuses on the query rather than merely describing the video. Figures 11 and 12 further showcase additional successful examples of our method.
>
> **[W3] Minor Fix**
>
> We thank the reviewer for the comment.  We will revise the manuscript to avoid implying many‐anchor advantages.

---

### Author Response · Authors · 2025-12-01
**Author Final Remarks**

We thank the AC and reviewers for their constructive feedback and time. We are encouraged that **Reviewers EJ9d and PYVz** recognized our core innovation, the **Progressive Anchor mechanism**, as positive and promising. They highlighted its capability to enhance reasoning accuracy, faithfulness, and interpretability by incorporating verifiable timestamp anchors into reasoning traces and enabling progressive coarse-to-fine refinement.

In the rebuttal, we have systematically addressed all concerns:

* **Computational Cost (EJ9d, PYVz):** We provided detailed hardware specifications (A100 40G GPUs) consistent with our implementation details and demonstrated that resolving specific hardware bottlenecks would halve the training time, addressing concerns regarding efficiency.

* **Generalization (PYVz, EVwv):** While our paper already demonstrates strong performance on TVG tasks, we provided new experiments in the rebuttal showing that our model, trained solely on TVG, achieves zero-shot improvements on VQA tasks. This further validates the generalization capability of our RL training strategy beyond the primary task.

* **Faithfulness & Interpretability (EJ9d, PYVz):** We added extensive visualization cases in the Appendix confirming the alignment between our reasoning traces and video content. Furthermore, we validated the faithfulness  of our reasoning process through a new counterfactual substitution experiment (PYVz), proving that our model relies on genuine visual evidence.

* **Method Clarification (PYVz, jaMU, EVwv):** We clarified the **RL-SFT-RL pipeline**. We specified that the initial RL stage is used *only* to curate the 30K CoT dataset by training a 7B model with RL. We emphasize that the model from the initial RL stage is not used for subsequent training. The subsequent SFT-RL stage follows standard paradigms.

* **Ablation Studies (EJ9d, PYVz):** We provided additional studies on more RL/Data/Backbone sizes (EJ9d), and the application of our prompt to stronger models (PYVz) to demonstrate robustness.

* **Presentation (jaMU, EVwv):** We have improved the manuscript's presentation by removing underlines in the references (jaMU) and increasing the font size in Figures 1 and 3 for better readability (EVwv).

We believe these revisions and additional experiments strongly support the validity and contribution of **TAR-TVG**.

---

### Meta-Review · Area_Chair_W5ur · 2026-01-07

**Summary:**

The paper proposes TAR-TVG, a method enhancing Large Vision-Language Models (LVLMs) for Temporal Video Grounding (TVG). The core approach involves a three-stage training pipeline (RL -> SFT -> RL) and a novel reward mechanism that enforces timestamp anchors within the reasoning chain, incentivizing progressive refinement of temporal predictions.

While the reviewers acknowledged the strong empirical performance (SOTA on Charades-STA and QVHighlights) and the intuitive motivation of explicit format constraints, there are still several major concerns as follows:

The primary contribution lies in the Progressive Refinement Reward, the reviewers argue this specific design is reasonable for the TVG task, but maybe not for other tasks. During discussion, the authors provide new experiments (VQA), which partially solve this issue.

The proposed training pipeline is heavily resource-intensive, which has also been clarified in the rebuttal, but this is still a concern regarding the performance gains.

A critical concern shared by reviewers is whether the reasoning is genuine. The proposed method constrains the format (XML tags) and the numerical accuracy of the timestamps, but it imposes no semantic constraints on the textual content. Consequently, there is no guarantee that the model isn't simply "babbling" or hallucinating plausible-sounding text to fill the gap between two timestamp predictions. The model may optimize for the reward by learning the structural template without actual visual reasoning logic. The "reasoning-enhanced" claim remains superficial if the textual reasoning itself is not verified for quality and causal link to the answer.

**Reviewer Concerns:**

see metareview

**Reviewer Scores:**

see metareview

---

### Decision · Program_Chairs · 2026-01-26

Reject